# A Blended Cognitive–Behavioral Intervention for the Treatment of Postpartum Depression: Study Protocol for a Randomized Controlled Trial

**DOI:** 10.3390/ijerph17228631

**Published:** 2020-11-20

**Authors:** Mariana Branquinho, Maria Cristina Canavarro, Ana Fonseca

**Affiliations:** Center for Research in Neuropsychology and Cognitive Behavioral Intervention, Faculty of Psychology and Educational Sciences, University of Coimbra, Rua do Colégio Novo, 3000-115 Coimbra, Portugal; mccanavarro@fpce.uc.pt (M.C.C.); ana.fonseca77@gmail.com (A.F.)

**Keywords:** postpartum depression, cognitive–behavioral therapy, blended treatment, Be a Mom, study protocol

## Abstract

Despite the existence of effective treatment for postpartum depression, few women seek professional help, indicating the need for a new and innovative format of treatment that can overcome help-seeking barriers. This article presents the study protocol for a blended cognitive–behavioral intervention for the treatment of postpartum depression, by integrating face-to-face sessions with a web-based program (Be a Mom) into one treatment protocol. This study will be a two-arm, noninferiority randomized controlled trial comparing blended intervention to usual treatment for postpartum depression provided in healthcare centers. Portuguese postpartum adult women diagnosed with postpartum depression (according to the DSM-5 diagnostic criteria for major depressive disorder) will be recruited during routine care appointments in local healthcare centers and will be eligible to participate. Measures will be completed at baseline, postintervention, and at three- and six-month follow-ups. The primary outcome will be depressive symptoms. Secondary outcomes will include anxiety symptoms, fatigue, quality of life, marital satisfaction, maternal self-efficacy, and mother–child bonding. Cost-effectiveness analysis and mediator and moderator analysis will be conducted. This study will provide insight into the efficacy and cost-effectiveness of a blended psychological intervention in the Portuguese context and increase the empirically validated treatment options for postpartum depression.

## 1. Introduction

Postpartum depression (PPD) is a serious clinical condition affecting approximately 13% of Portuguese women after childbirth [1]. When left untreated, PPD poses adverse and persistent consequences for the entire family system. It affects the woman’s health (e.g., increased tiredness [2], decreased quality of life [3]) and mother–child interaction (e.g., mother-child bonding, lower parenting self-efficacy) [3,4]. Moreover, it can have consequences for the infant’s development (e.g., infant sleep patterns, emotional development) [3] and for the entire family environment, including the couple’s relationship [5].

Despite the existence of effective treatments (e.g., cognitive–behavioral therapy [CBT]) [6], few women with PPD seek professional help [7]. A Portuguese study revealed that only 13.6% of women with depressive symptoms during the perinatal period sought professional help to address their emotional difficulties [8]. Time and financial constraints and struggles with transportation and childcare issues are some of the structural barriers to seeking professional help reported by postpartum women [9,10], suggesting the need for new delivery formats to improve women’s access to evidence-based PPD interventions.

E-mental health tools are an innovative form of treatment delivery that use digital technology, including web-based technology, in the mental health field [11]. These tools can overcome PPD treatment uptake barriers given their reduced costs, flexibility, and improved accessibility [12]. Women in the postpartum period already use the internet frequently to search for information about PPD [10]. Moreover, e-mental health tools have been perceived as acceptable and useful among Portuguese women in the perinatal period, particularly among those women presenting clinically relevant depressive symptoms [13].

Existing web-based interventions for PPD treatment based on CBT have proven to be effective in the reduction of postpartum depressive symptoms [14,15,16]. Interventions such as MomMoodBooster [17], NetMums [18], and Mom-Net [19] have shown promising results not only in reducing postpartum depressive symptoms but also in improving self-efficacy, marital relationship, and mother–child bonding.

However, there is also evidence that web-based interventions suffer from important limitations related to the accuracy of diagnosis, which is based only on online assessments [20], and with low engagement and high attrition rates [15,16] due to the absence of therapist support during the intervention [15,20]. Web-based interventions also lack nonverbal communication as well as the opportunity to discuss specific problems and to deal with crises [21,22]. Instead of replacing traditional psychological interventions, e-mental health tools can be an important complement to them [23].

Blended treatment is the combination of face-to-face treatment with web-based interventions that are integrated and used sequentially in one treatment protocol [24]. Therefore, delivering PPD treatment using a blended format could benefit from the potential of both treatment modalities (face-to-face and online) [23]. Blended treatment presents the advantages of the utilization of e-mental health tools, namely, flexibility in application, good accessibility, and travel time savings [21,25]. Additionally, online sessions can improve patient self-management and help patients better prepare for a session with a therapist [21,22]. Blended treatment allows professional guidance in the therapeutic process, which increases adherence, prevents dropout, facilitates increased treatment intensity, and leads to better results compared to unguided treatments [20]. CBT therapists recognize that blended intervention formats support the patient’s motivation, can be adjusted to the patient’s specific needs, and reduce the treatment gap between sessions [23,26]. Online sessions can also replace some face-to-face sessions with the therapist, allowing for time savings in healthcare systems as well as decreased treatment costs [24,27].

There is growing evidence of the efficacy of blended treatments for several psychological disorders [24], including depression [21,28,29]. Existing studies have indicated that blended treatment for depression is perceived positively by patients [22,28]. Despite its advantages and considering the aforementioned barriers to professional help-seeking in the postpartum period [9,10], to our knowledge, there is no blended treatment format targeting PPD.

This article presents the study protocol for a blended CBT intervention combining face-to-face sessions with the online program Be a Mom for the treatment of PPD in the Portuguese context. In Portugal, the Be a Mom program was developed as a culturally sensitive web-based CBT intervention that is designed as a self-guided tool for the prevention of PPD. Preliminary evidence of Be a Mom’s pilot trial suggests its effectiveness in reducing depressive symptoms among women presenting early-onset PPD symptoms [30], thus supporting its potential as a PPD treatment tool integrated into a blended treatment protocol.

Therefore, we herein outline the protocol for a randomized controlled trial to examine the acceptability and efficacy of a blended CBT intervention for PPD treatment, considering postintervention and follow-up improvements in primary and secondary outcomes. It is expected that the blended CBT intervention will be as effective as treatment usually provided for PPD in decreasing depressive symptoms. In this study, we will evaluate the mediating role of psychological competences (self-compassion, emotion regulation, psychological flexibility) in treatment response. These mechanisms have been core psychological processes underlying the development of the Be a Mom program [31]. Moreover, previous studies have found that these psychological mechanisms were associated with improvements in depressive symptoms in the perinatal period [30,32,33]. We will also examine the moderator effect of characteristics of the patient (e.g., sociodemographic characteristics, motivation for therapy) and of the therapeutic process (e.g., therapeutic relationship, user’s satisfaction) in the efficacy of the blended intervention for PPD.

## 2. Materials and Methods

### 2.1. Study Design

This study will be a two-arm, noninferiority randomized controlled trial (RCT) comparing blended CBT intervention for PPD (Blended Be a Mom) to the usual treatment that women receive to treat PPD in primary healthcare centers (treatment as usual; TAU). Participants in both the Blended Be a Mom and TAU conditions will complete baseline, postintervention, and follow-up (three and six months postintervention) assessments through a link sent by email that gives access to the survey.

### 2.2. Ethical Issues

This study was approved by the Ethics Committee of the Faculty of Psychology and Educational Sciences, University of Coimbra, and it will follow the ethical standards and procedures for research with human beings [34,35]. This study protocol was registered with ClinicalTrial.gov (Protocol Record NCT04441879). Participants will be informed about the study goals and procedures and the researcher and participants’ roles. An informed consent form to participate in the study will be signed by participants. Participation in the study will be free of cost to women, and no compensation will be given. Women can withdraw at any time, and dropout will not compromise medical care. All collected data will be stored in a secure server in accordance with the General Data Protection Regulation (GDPR) and will only be used for the purposes of the present study. Participants’ information will be confidential and anonymized (i.e., no personal data that allow the participant’s identification) and will only be treated at a collective level. Trial results will be shared with both the scientific community and health professionals, through publications in scientific peer-reviewed journals and presentations at national and international conferences.

### 2.3. Participants (Inclusion and Exclusion Criteria)

Adult women during the postpartum period (up to 12 months postpartum) with a confirmed diagnosis of PPD (according to the Structured Clinical Interview for DSM-5 [SCID-5] disorders criteria) will be eligible to participate in this study. Additionally, participants must be residents of Portugal, be able to write and read Portuguese, and have regular access to computers and the internet.

Exclusion criteria will include the presence of psychiatric comorbidity requiring alternative treatment primary to depression treatment, the presence of suicidal ideation, a serious medical condition of either the mother or the baby, and current treatment for depression (e.g., other psychological interventions). Participants who are not eligible to participate in the study will be referred to intervention by local providers.

### 2.4. Recruitment and Eligibility Assessment

Participants will be recruited in primary healthcare units of the region in routine care appointments during the postpartum period. Alternative recruitment methods (e.g., other institutions, online advertisement) will be considered if sample recruitment difficulties arise (e.g., if the sample size is not achieved, or if the current COVID-19 pandemic disrupts the contact with patients within healthcare institutions). Local healthcare providers (e.g., primary care nurses) will be informed about the study and will ask women if they are interested in participating. Women will be informed in detail about the study, both verbally and through a written flyer. If they are willing to participate, they will be asked to sign an informed consent form and to complete a questionnaire including sociodemographic information, a questionnaire to screen for the presence of depressive symptoms (Edinburgh Postnatal Depression Scale), and other eligibility criteria questions (e.g., technology access, not currently undergoing treatment for PPD). Assessment of depressive symptoms will be conducted every two weeks during the period of the study. When women have a positive screen (indicating the presence of clinically relevant depressive symptoms) and meet the remaining eligibility criteria, they will be further contacted by the researchers through telephone or email to inform them that they will proceed to the second phase of the study. In the second phase, an interview (SCID-5) will be conducted by the researcher (licensed psychologist) to assess the presence of the diagnosis of PPD. Women with a clinical diagnosis of PPD will be eligible to participate in the study and will be included in the third phase of the study. In the third phase of the study, eligible women will receive an email containing a link to complete an online self-report questionnaire (baseline assessment). The flowchart of the study is presented in Figure 1, demonstrating the recruitment and eligibility assessment.

### 2.5. Randomization

After completing the baseline assessment, participants will be randomly assigned (blocked randomization, allocation 1:1) to the intervention (Blended Be a Mom) or the TAU conditions (see Figure 1). Randomization will be conducted by a researcher blind to the assessment procedure and will be performed using a computerized random number generator. Women in both conditions will be informed about their assigned treatment condition. Blinding for treatment conditions will not be possible.

### 2.6. Interventions

#### 2.6.1. Blended Intervention

The blended protocol will be developed based on existing evidence-based CBT interventions for PPD delivered both face-to-face (e.g., [36]) and online (e.g., [37]). The final blended CBT intervention protocol for PPD (Blended Be a Mom) will be developed by the research team and reviewed and approved by a panel of researchers with clinical expertise in the area of PPD. A pilot study with women with a clinical diagnosis of PPD will be conducted prior to the RCT, to assess the acceptability and feasibility of the structure and content of the blended intervention, and to gather preliminary evidence of its clinical efficacy (noncontrolled). Appropriate adjustments to the blended intervention protocol will be done accordingly.

The Blended Be a Mom intervention will integrate 7 face-to-face CBT sessions that are weekly alternated with 6 online sessions over a period of 13 weeks. The online part of the blended intervention will be adapted from the Be a Mom program. Both face-to-face and online sessions will be designed according to CBT principles: problem-oriented, structured, time-limited, educative, and promoting the active participation of the patient [38]. The content of sessions will include psychoeducation, cognitive strategies for negative thoughts, behavioral activation, and relapse prevention (more detailed information is presented in Figure 2). The intervention will also include the utilization of a mobile phone application to conduct ecological momentary assessment, a method to collect information in real time and in the natural environment of participants, over a period of time [39].

Each face-to-face session (with an approximate length of 45–60 min) begins with mood checking and discussion of women’s symptoms. The therapist then reviews the experience with the online program and each module’s content (i.e., to discuss homework assignments and to practice the strategies learned in the online session), provides feedback, and discusses any doubts. The session ends with the presentation of the upcoming online program module’s objectives. The face-to-face sessions will be delivered by a predoctoral-level licensed psychologist, with the supervision of an experienced postdoctoral-level psychologist. To ensure fidelity to treatment protocol, a detailed therapist manual will be available and weekly supervision will be provided by a senior psychologist. At the end of each session, the therapist will fill a checklist to confirm that the topics of the session were covered.

The Be a Mom program was originally designed for the prevention of PPD among Portuguese women. It contains five modules addressing several thematic contents (e.g., Changes and Emotional Reactions, Managing Negative Thoughts, Values and Social Support) and incorporates the recent contributions of third-wave CBT approaches (e.g., self-compassion and acceptance and commitment therapy). Adaptation will be made to the modules to address the specific needs of PPD intervention. Each online session (with an approximate length of 30–45 min) opens with an introduction to the session goals and content, followed by specific information and strategies. Exercises and activities are included to practice the session’s specific content, and information is presented through different formats, such as text, interactions, animation, and videos.

After participants access the program, all modules will be available. Participants will be instructed to complete one module at a time (one session per week) alternating with face-to-face sessions. Online sessions (Be a Mom modules) will be self-guided and an asynchronous communication channel with the therapist through the program will be available. Before entering a new module, participants must confirm that it is in accordance with the therapist. Participants can pause the module at any time and resume the last page visited during subsequent access. Email reminders will be sent to participants to motivate and encourage engagement in online sessions.

The blended intervention will be discontinued if there is a high risk for suicide, possibility to harm others, or the development of severe depressive symptoms. Risk assessment during the intervention, postintervention, and at follow-up assessments will be conducted, through both self-reported and EPDS scores and specific suicidal intention item on the questionnaires that will be administered. These participants will be immediately referred to other mental health services (psychological or psychiatric services) and their participation in the blended intervention will end.

#### 2.6.2. Treatment as Usual

TAU involves the treatment provided in routine healthcare for PPD. It can include different types of traditional face-to-face treatment (e.g., CBT, interpersonal psychotherapy). TAU will be conducted by healthcare center providers (e.g., psychologists), and information concerning the type and duration of therapy (e.g., number of sessions) will be obtained.

### 2.7. Measures

Table 1 presents the study variables and assessment times.

#### 2.7.1. Sociodemographic, Clinical, and Obstetric Information

Women’s sociodemographic (e.g., age, marital status, number of children, educational level, professional status, average monthly income, socioeconomic status and residence) and obstetric (e.g., parity, pregnancy complications, type of labor, postpartum data) information will be collected through a questionnaire developed by the researchers. It will also include self-reported clinical information concerning history of psychological/psychiatric problems (“Have you had psychological or psychiatric problems [e.g., depression, anxiety]?”, Yes or No) and history of psychological/psychiatric treatment (“Have you had psychological or psychiatric treatment?”, Yes or No). Infant-related information (e.g., infant age, infant sex, infant gestational weeks at birth, infant feeding patterns) will also be collected.

#### 2.7.2. Primary Outcome

Changes in depressive symptoms, the primary outcome, will be measured with the Portuguese version of the Edinburgh Postnatal Depression Scale (EPDS) [40].The EPDS is a 10-item scale (e.g., “I have felt sad or miserable”) that assesses how women felt over the last seven days concerning several symptoms using an individualized four-point Likert scale (from 0 to 3). The total score can range between 0 and 30, and higher scores are indicative of more severe depressive symptoms. In Portuguese validation studies, a score of 10 or higher suggests the presence of clinically relevant depressive symptoms. The Portuguese version of EPDS showed good levels of internal consistency (Cronbach’s alpha = 0.85) and adequate validity [40].

#### 2.7.3. Secondary Outcomes

Anxiety symptoms will be measured with the Anxiety Subscale of the Portuguese version of the Hospital Anxiety and Depression Scale (HADS-A) [41]. This subscale comprises seven items (e.g., “Worrying thoughts go through my mind”) answered on a four-point response scale (ranging from 0 to 3). Higher scores indicate more symptomatology. A score of 11 or higher is indicative of the presence of clinically relevant anxiety symptoms. The Portuguese version of HADS [41] is a reliable scale, with an adequate internal consistency (Cronbach’s alpha = 0.76 for the Anxiety Subscale).

Fatigue will be measured with the Portuguese version of the Fatigue Severity Scale (FSS) [42]. The FSS is composed of nine items (e.g., “Fatigue interferes with my work, family, or social life”) answered on a seven-point scale ranging from 1 (strongly disagree) to 7 (strongly agree). Higher scores suggest more severe fatigue. The Portuguese version of FSS proved to be a reliable and valid instrument, with a good internal consistency (Cronbach’s alpha = 0.87) [42].

Quality of life will be assessed with the Portuguese version of the Euroqol Five-Dimension Scale (EQ-5D) [43]. It is composed of five items (mobility, self-care, usual activities, pain/discomfort, and anxiety/depression), and each item is rated on a scale ranging from 1 (no problems) to 3 (extreme problems). Additionally, participants are asked to rate their own health through visual analogue on a scale ranging from 0 (worst imaginable health state) to 100 (best imaginable health state). The total score is obtained through an algorithm (the digits of the answers to five dimensions) and describes the health state. The Portuguese version of EQ-5D has adequate levels of internal consistency (Cronbach’s alpha = 0.72) and was found to be a valid and reliable measure [43].

Marital satisfaction will be assessed with the Portuguese version of the Investment Model Scale—Satisfaction subscale (IMS) [44]. This subscale comprises five items (e.g., “My relationship is close to ideal”) rated on a nine-point scale ranging from 0 (do not agree at all) to 8 (completely agree). Higher scores suggest higher satisfaction with the relationship. The Portuguese version of IMS presented good reliability and validity, and found a Cronbach’s alpha of 0.91 for the Satisfaction subscale [44].

Maternal self-efficacy will be assessed with the Portuguese version of the Perceived Maternal Parenting Self-Efficacy Questionnaire (PMPS-E; psychometric studies ongoing) [45]. This instrument comprises 20 items (e.g., “I can read my baby’s cues”) answered on a four-point scale ranging from 1 (strongly disagree) to 4 (strongly agree). Higher scores are indicative of higher perceived maternal self-efficacy.

Mother–child bonding will be measured with the Portuguese version of the Postpartum Bonding Questionnaire (PBQ) [46]. The PBQ is a 12-item instrument (e.g., “I feel close to my baby”) with a six-point Likert answer scale ranging from 0 (never) to 5 (always). Higher scores are indicative of a more impaired mother–child bond. The Portuguese version of PBQ found good levels of internal consistency (Cronbach’s alpha = 0.71) and validity [46].

Ecological momentary assessments of mood (rated on a scale from “very low” to “very good”), self-esteem, motivation, ability to feel pleasure, depressed mood, and insomnia (Yes or No) will be conducted on a daily basis.

#### 2.7.4. Psychological Competences

Emotion regulation difficulties will be assessed with the Portuguese version of the Difficulties in Emotion Regulation Scale—Short Form (DERS-SF; psychometric studies ongoing) [47]. The DERS-SF is a self-report instrument composed of 18 items (e.g., “When I’m upset, I believe there is nothing I can do to make myself feel better”) answered on a five-point scale ranging from 1 (almost never) to 5 (almost always). Higher scores indicate more difficulties in emotion regulation.

Self-compassion will be measured with the Portuguese version of the Self-Compassion Scale—Short Form (SCS-SF) [48]. This is a 12-item instrument (e.g., “I try to see my failings as part of the human condition”) with a five-point response scale ranging from 1 (almost never) to 5 (almost always). Higher scores suggest higher levels of self-compassion. The Portuguese version of SCS-SF is a valid and reliable instrument, with good internal consistency (Cronbach’s alpha = 0.86) [48].

Psychological flexibility will be assessed with the Portuguese version of the Acceptance and Action Questionnaire-II (AAQ-II) [49]. The AAQ-II comprises seven items (e.g., “I’m afraid of my feelings”) rated on a seven-point scale ranging from 1 (never true) to 7 (always true). Higher scores are indicative of lower psychological flexibility (i.e., higher psychological inflexibility). The Portuguese version of AAQ-II showed good internal consistency (Cronbach’s alpha = 0.90) and adequate validity [49].

#### 2.7.5. Intervention-Related Outcomes

Motivation for therapy will be measured with the Portuguese version of the Client Motivation for Therapy Scale (CMTS; psychometric studies ongoing) [50]. The CMTS comprises 24 items (e.g., “Because I would like to make changes to my current situation”) rated on a seven-point response scale ranging from 1 (not true at all) to 7 (totally true). Higher scores suggest higher motivation for therapy. 

The therapeutic relationship will be assessed with the Portuguese version of the Working Alliance Inventory—Short revised (WAI-SR) [51], a 12-item instrument with a five-point response scale ranging from 1 (rarely or never) to 5 (always). Higher scores indicate better therapeutic alliance. The Portuguese version of WAI-SR is a reliable measure and has good levels of internal consistency (Cronbach’s alpha = 0.85).

Acceptability, satisfaction, and usability of the blended treatment will be measured through specific questions developed by the researchers (e.g., satisfaction with the program, usefulness, acceptability, demandingness, recruitment rate, dropout rate, web system data).

The feasibility of the program will be assessed through website utilization (e.g., number of logins, average visit length, total time spent on the website, number of exercises completed) and dropout rate.

#### 2.7.6. Economic Evaluation

Cost-effectiveness will be assessed with an adapted version of the Treatment Inventory Cost in Psychiatric Patients (TiC-P) [52]. This instrument measures medical costs and indirect nonmedical costs, through the assessment of the participant’s healthcare use in the last three months (i.e., the number of contacts with healthcare providers), productivity losses (i.e., the number of days of absence from work due to illness), and efficiency at work in the last four weeks.

### 2.8. Sample Size and Statistical Analyses

The sample size for this study was determined based on power analysis (G*Power). A sample of 45 women per condition is required to detect medium effects in comparison analyses, considering the primary outcome. Considering an expected dropout rate of 20%, we plan to recruit a sample of at least 110 participants (55 per condition) to account for attrition effects.

Statistical analyses to examine the efficacy of the program will be conducted following the intention-to-treat (ITT) and per-protocol (PP) principles in accordance with the CONSORT recommendations [53]. ITT analyses allow us to examine data from all randomized participants, even those with missing values on outcome measures. In contrast, PP analysis includes only participants who followed the assigned treatment protocol. Statistical analyses will be performed using the Statistical Package for the Social Sciences (SPSS, Version 25.0; IBM SPSS) and the Mplus program (Version 7). Linear mixed models will be conducted to determine the effects of the intervention over time (time × group interaction effects) on primary and secondary outcomes and changes in psychological competences. Other appropriate statistical analyses such as two-wave latent change score models, reliable change index, Chi-square tests, and within-group effect sizes will be performed, as well as mediator and moderator analysis. Preliminary cost-effectiveness analysis will be conducted from a healthcare cost perspective and a societal perspective, comparing the differences between the intervention and control group on the outcomes, with the differences in the costs generated. The contribution of a health economics expert will be required.

## 3. Discussion

Despite the existence of effective treatment, few women seek professional help to deal with their depressive symptoms in the postpartum period [7,8], indicating the need for a new and innovative format of treatment that can overcome help-seeking barriers. This study aims to evaluate the acceptability and effectiveness of a blended CBT intervention for the treatment of PPD in the Portuguese context by integrating face-to-face sessions with the web-based program Be a Mom.

To our knowledge, this will be the first study to develop a blended CBT treatment protocol for PPD. Blended Be a Mom benefits from both treatment formats, offering the flexibility, accessibility, and self-management of e-health tools as well as clinical support, increased motivation, and higher treatment intensity [21]. Moreover, it can potentially decrease the number of face-to-face sessions and reduce costs in healthcare systems. A blended CBT intervention can therefore increase help-seeking behaviors among women in the postpartum period by providing treatment that mitigates the impact of the identified barriers in professional help-seeking.

Existing studies have revealed that blended treatment for depression can be effective in reducing depressive symptoms and maintaining these gains over a period of six months [28]. Additionally, previous findings have shown that blended interventions are more effective compared to control groups without intervention (i.e., waiting lists) [24] and that it can be as effective as standard CBT treatments [29]. We expect that Blended Be a Mom will be as effective as TAU with regard to long-term effects on primary and secondary outcomes. The feasibility, acceptability, and usability of the blended intervention will be considered in addition to its cost-effectiveness.

Despite its advantages, there is still limited knowledge about the suitability of blended treatment for every patient [20]. Characteristics such as age, severity of symptoms, or the ability to use technology should be considered and further studied to optimize the effectiveness of blended interventions. The results of our study will provide insights into the processes underlying the treatment effects of blended intervention and the characteristics that moderate the effectiveness of blended intervention.

Our study will also be innovative due to the inclusion of daily ecological momentary assessments during the intervention. This approach allows the collection of information in women’s natural environment in real time and therefore prevents retrospective biases [39]. This will provide important information about intraindividual variations over the treatment, the dynamic evolution of PPD symptoms over time, and temporal relationships between mood and other experiences. The inclusion of ecological momentary assessment is recommended in RCTs because it can optimize statistical power effects, improve measurements precision, and potentially increase treatment’s adherence [54]. This data collection method has previously been used both in the postpartum period and in depression disorders, and it was considered feasible and acceptable by the users [55,56].

## 4. Conclusions

This will be the first study to develop and test the effectiveness of a blended CBT intervention for the treatment of PPD in the Portuguese context. This innovative format of treatment delivery can potentially reduce costs in healthcare systems, increase its efficiency, and promote help-seeking behaviors among women in the postpartum period.

We will contribute to the existing research on the topic of e-health technologies applied to mental health. This study is in line with the current directions from the Portuguese e-health Strategy [57] and the European e-Health Action Plan 2012–2020 [58] that encourage the integration of web-based technologies into clinical practice and the use of these tools to enhance patient-centered care and to increase health systems’ sustainability and efficiency. We will provide the Portuguese population with access to an evidence-based blended psychological intervention for PPD treatment while contributing to the more effective management of resources in healthcare services.

## Figures and Tables

**Figure 1 ijerph-17-08631-f001:**
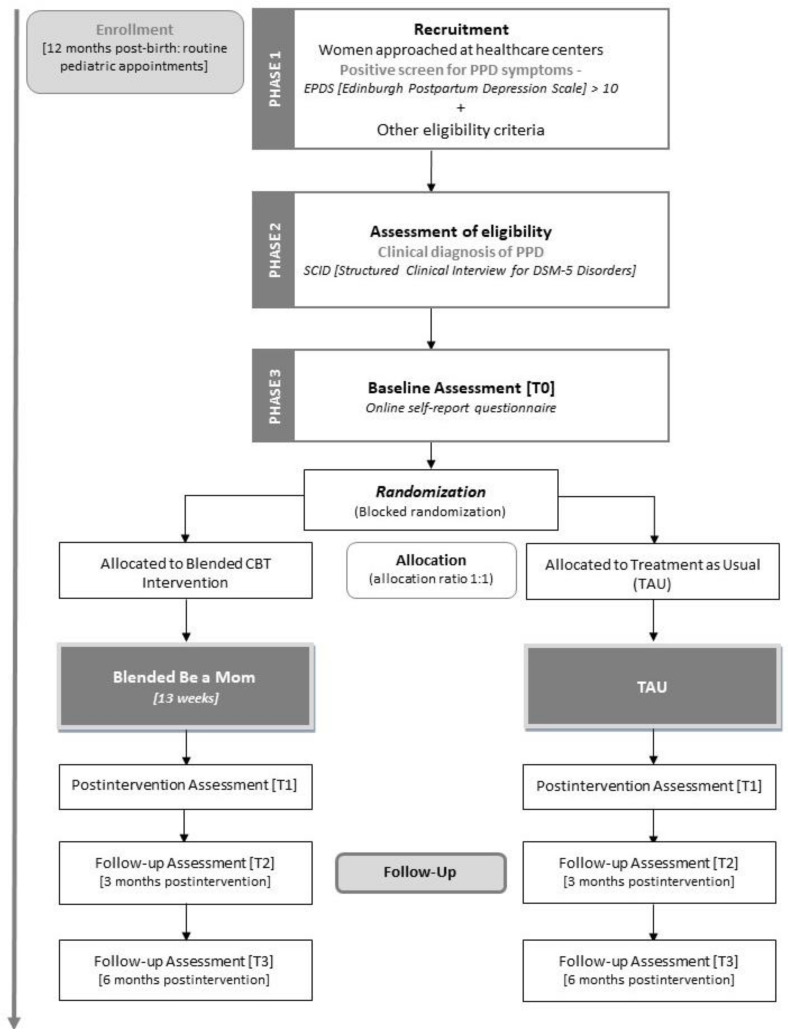
Flowchart of the study.

**Figure 2 ijerph-17-08631-f002:**
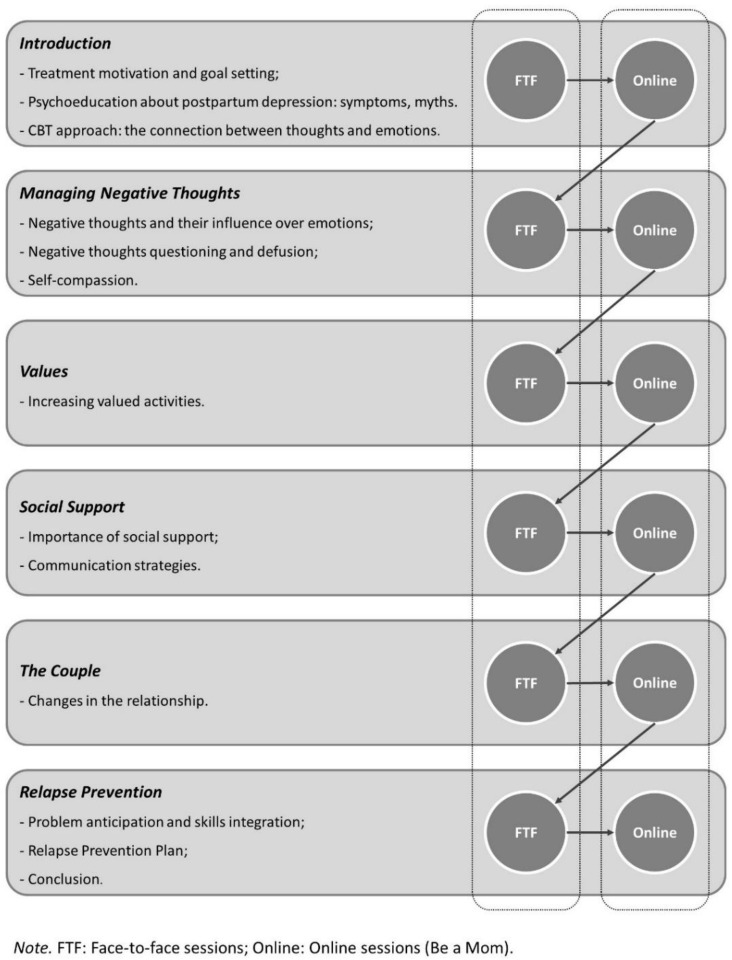
Blended Be a Mom—structure and content of sessions.

**Table 1 ijerph-17-08631-t001:** Study variables and assessment points.

Variables	Baseline [T0]	Postintervention [T1]	Follow-Up [T2]	Follow-Up [T3]
Sociodemographic, clinical and obstetric information	x			
Depressive symptoms	x	x	x	x
Anxiety symptoms	x	x	x	x
Fatigue	x	x	x	x
Quality of life	x	x	x	x
Marital satisfaction	x	x	x	x
Maternal self-efficacy	x	x	x	x
Mother–child bonding	x	x	x	x
Self-compassion	x	x		
Emotion regulation	x	x		
Psychological flexibility	x	x		
Motivation for therapy	x	x		
Therapeutic relationship		x		
Acceptability, satisfaction, and usability		x		
Economic evaluation	x	x

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
