# Peer review of "A Blended Cognitive–Behavioral Intervention for the Treatment of Postpartum Depression: Study Protocol for a Randomized Controlled Trial"

_ijerph, 2020, doi:10.3390/ijerph17228631_

Round 1
Reviewer 1 Report
This is a protocol for a non-inferiority, randomized controlled trial testing the feasibility/acceptability and preliminary efficacy of a blended CBT intervention for postpartum depression. The protocol is very well-written. The topic is relevant to the journal’s scope and addresses a topic of high public health impact. The introduction is comprehensive and clear. Because this is a pilot trial, I would encourage the authors to use “preliminary efficacy” rather than effectiveness. Below, I indicate several areas that would benefit from clarification and more detailed presentation.
Introduction
The introduction would benefit from a more explicit mention of evidence related to the chosen secondary outcomes and competences. It would allow for a stronger rationale for the selection of these variables.
Importantly, please indicate the hypotheses and research questions of your proposed study.
Sampling and inclusion criteria
Elaborate on any strategies used for achieving adequate participant enrollment. How do you anticipate to reach target sample size?
Presence of severe symptoms warranting psychiatric care will be an exclusion. However, there is no discussion of crisis management plan, should a participant worsen or become suicidal during the intervention. What will the criteria for discontinuing or modifying the proposed intervention and what will the plan be?
Proposed intervention and therapists
More information on who will perform he F2F sessions. Qualifications, training, experience with blended forms of therapy. How many therapists will be participating in the trial?
Will there be an assessment of therapist adherence/fidelity to protocol?
It is not clear to me if online sessions are self-guided and asynchronous (I would assume they are, but I think it should be made clear). Could you please clarify?
Measures
The authors discuss the use of ecological momentary assessment to collect information in real time and in the natural environment of participants. Could they discuss this further and explain how they will use this?
Regarding the economic evaluation of the proposed intervention, the authors will employ the Treatment Inventory Cost in Psychiatric Patients instrument. A more detailed description of the tool would be helpful, along with a more detailed strategy of how cost-effectiveness will be evaluated.
Data analytic plan
For sample and statistical analysis, which data analytic plan will the authors employ to test their primary hypotheses?
It is mentioned in the abstract that “mediator and moderator analysis will be conducted” but there is no discussion of this further in the main text. Which variables will the authors investigate as moderators and mediators?
Minor edits
Line 306, please change to “existing”
Author Response
Response to Reviewer 1 Comments
Point 1: This is a protocol for a non-inferiority, randomized controlled trial testing the feasibility/acceptability and preliminary efficacy of a blended CBT intervention for postpartum depression. The protocol is very well-written. The topic is relevant to the journal’s scope and addresses a topic of high public health impact. The introduction is comprehensive and clear. Because this is a pilot trial, I would encourage the authors to use “preliminary efficacy” rather than effectiveness. Below, I indicate several areas that would benefit from clarification and more detailed presentation.
Response 1: Thank you for your positive comments and your suggestions. Concerning your comment, a pilot study will be previously conducted to test the intervention’s preliminary efficacy. If necessary, adaptations to the protocol will be made and then the randomized controlled trial will be conducted. We agree that this information is not clear in the manuscript, so we added this information, it now reads:
“A pilot study with women with a clinical diagnosis of PPD will be conducted prior to the RCT, to assess the acceptability and feasibility of the structure and content of the blended intervention, and to gather preliminary evidence of its clinical efficacy (non-controlled). Appropriate adjustments to the blended intervention protocol will be done accordingly.” (see page 4)
Introduction
Point 2: The introduction would benefit from a more explicit mention of evidence related to the chosen secondary outcomes and competences. It would allow for a stronger rationale for the selection of these variables.
Response 2: Thank you for your suggestion. The choice of the secondary outcomes of this study is justified by two main arguments: 1) secondary outcomes that are related with the impact of PPD; 2) secondary outcomes that constitute psychological processes/ skills that are modifiable and that we believe can act as mechanisms of treatment response. Concerning the first set of secondary outcomes, we present in the first paragraph of the introduction the literature evidence of their association with PPD:
“It affects the woman’s health (e.g., increased tiredness [2], decreased quality of life [3]) and mother-child interaction (e.g., mother-child bonding, lower parenting self-efficacy) [3, 4].”
Also, we show that existent interventions have proved to be effective in some of these secondary outcomes:
“Interventions such as MomMoodBooster [17], NetMums [18] and Mom-Net [19] have shown promising results not only in reducing postpartum depressive symptoms but also in improving self-efficacy, marital relationship and mother-child bonding.” Therefore, we believe that repeating this information at the end of the information will not significantly add to the paper and will limit the flow of the reading.
However, concerning the second set of secondary outcomes (psychological processes/competences), we agree with the reviewer that the choice to evaluate the mentioned psychological competences is not clear, and we added the information that previous interventions have found significant associations between these variables and postpartum depression. It now reads:
“In this study, we will evaluate the mediating role of psychological competences (self-compassion, emotion regulation, psychological flexibility) in treatment response. These mechanisms have been core psychological processes underlying the development of the Be a Mom program [31]. Moreover, previous studies have found that these psychological mechanisms were associated with improvements in depressive symptoms in the perinatal period [30, 32, 33].” (see pages 2 - 3)
Point 3: Importantly, please indicate the hypotheses and research questions of your proposed study.
Response 3: We added the following paragraph:
“Therefore, we herein outline the protocol for a randomized controlled trial to examine the acceptability and efficacy of a blended CBT intervention for PPD treatment, considering post-intervention and follow-up improvements in primary and secondary outcomes. It is expected that the blended CBT intervention will be as effective as treatment usually provided for PPD in decreasing depressive symptoms. In this study, we will evaluate the mediating role of psychological competences (self-compassion, emotion regulation, psychological flexibility) in treatment response. These mechanisms have been core psychological processes underlying the development of the Be a Mom program [31]. Moreover, previous studies have found that these psychological mechanisms were associated with improvements in depressive symptoms in the perinatal period [30, 32, 33]. We will also examine the moderator effect of characteristics of the patient (e.g., sociodemographic characteristics, motivation for therapy) and of the therapeutic process (e.g., therapeutic relationship, user’s satisfaction) in the efficacy of the blended intervention for PPD.” (see pages 2 - 3)
Sampling and inclusion criteria
Point 4: Elaborate on any strategies used for achieving adequate participant enrollment. How do you anticipate to reach target sample size?
Response 4: Women will be approached at routine care appointments that they should attend after the baby is born. If the sample size is not achieved by recruitment at healthcare centers, or if the current covid-19 pandemic disrupts the contact with patients within healthcare institutions, alternative recruitment methods (e.g., other institutions, online advertisement) will be considered. We added this information on main text, it now reads:
“Alternative recruitment methods (e.g., other institutions, online advertisement) will be considered if sample recruitment difficulties arise (e.g., if the sample size is not achieved, or if the current covid-19 pandemic disrupts the contact with patients within healthcare institutions).” (see page 3)
Point 5: Presence of severe symptoms warranting psychiatric care will be an exclusion. However, there is no discussion of crisis management plan, should a participant worsen or become suicidal during the intervention. What will the criteria for discontinuing or modifying the proposed intervention and what will the plan be?
Response 5: As suggested, we added information concerning risk management and the criteria for discontinuing the intervention. It reads:
“The blended intervention will be discontinued if there is a high risk for suicide, possibility to harm others or the development of severe depressive symptoms. Risk assessment during the intervention, post-intervention and at follow-up assessments will be conducted, through both self-reported and EPDS scores and specific suicidal intention item on the questionnaires that will be administered. These participants will be immediately referred to other mental health services (psychological or psychiatric services) and their participation in the blended intervention will end.” (see page 7)
Proposed intervention and therapists
Point 6: More information on who will perform he F2F sessions. Qualifications, training, experience with blended forms of therapy. How many therapists will be participating in the trial?
Response 6: As suggested, we added the following information about the person who will perform the presential sessions:
“The face-to-face sessions will be delivered by a pre-doctoral level licensed psychologist, with the supervision of an experienced post-doctoral level psychologist.” (see page 6)
Point 7: Will there be an assessment of therapist adherence/fidelity to protocol?
Response 7: The researcher will follow a detailed therapist manual and will have regular (weekly) supervision by an expert psychologist in the field of postpartum depression, to discuss the previous session and to prepare the following sessions. At the end of each session, the therapist will fill a checklist to confirm that the topics of the session were addressed. We added the following information:
“To ensure fidelity to treatment protocol, a detailed therapist manual will be available and weekly supervision will be provided by a senior psychologist. At the end of each session, the therapist will fill a checklist to confirm that the topics of the session were covered.” (see page 6)
Point 8: It is not clear to me if online sessions are self-guided and asynchronous (I would assume they are, but I think it should be made clear). Could you please clarify?
Response 8: We presented the information that the modules of the Be a Mom program are self-guided (see page 2, lines 84-85). To make this clearer, we also added this information by the end of the section ‘Blended Intervention’. It now reads:
“Online sessions (Be a Mom modules) will be self-guided and an asynchronous communication channel with the therapist through the program will be available.” (see page 7)
Measures
Point 9: The authors discuss the use of ecological momentary assessment to collect information in real time and in the natural environment of participants. Could they discuss this further and explain how they will use this?
Response 9: More information about the use of ecological momentary assessment and how the information will be used was added in Discussion, it now reads:
“This will provide important information about intraindividual variations over the treatment, the dynamic evolution of PPD symptoms over time and temporal relationships between mood and other experiences. The inclusion of ecological momentary assessment is recommended in RCTs because it can optimize statistical power effects, improve measurements precision and potentially increase treatment’s adherence [54]. This data collection method has previously been used both in the postpartum period and in depression disorders, and it was considered feasible and acceptable by the users [55, 56].” (see page 10)
Point 10: Regarding the economic evaluation of the proposed intervention, the authors will employ the Treatment Inventory Cost in Psychiatric Patients instrument. A more detailed description of the tool would be helpful, along with a more detailed strategy of how cost-effectiveness will be evaluated.
Response 10: As suggested, we added more detailed information about the instrument, at ‘2.7.6. Economic evaluation’:
“This instrument measures medical costs and indirect nonmedical costs, through the assessment of the participant’s healthcare use in the last 3 months (i.e., the number of contacts with healthcare providers), productivity losses (i.e., the number of day of absence from work due to illness) and efficiency at work in the last 4 weeks.” (see page 9)
Also, the analysis about cost-effectiveness evaluation were clarified, it now reads:
“Preliminary cost-effectiveness analysis will be conducted from a healthcare cost perspective and a societal perspective, comparing the differences between the intervention and control group on the outcomes, with the differences in the costs generated. The contribution of a health economics expert will be required.” (see page 10)
Data analytic plan
Point 11: For sample and statistical analysis, which data analytic plan will the authors employ to test their primary hypotheses?
Response 11: We completed the section ‘Sample size and statistical analyses’, it now reads:
“Linear mixed models will be conducted to determine the effects of the intervention over time (time x group interaction effects) on primary and secondary outcomes and changes on psychological competences. Other appropriate statistical analyses such as two-wave latent change score models, reliable change index, Chi-square tests, and within-group effect sizes will be performed, as well as mediator and moderator analysis. Preliminary cost-effectiveness analysis will be conducted from a healthcare cost perspective and a societal perspective, comparing the differences between the intervention and control group on the outcomes, with the differences in the costs generated. The contribution of a health economics expert will be required.” (see page 10)
Point 12: It is mentioned in the abstract that “mediator and moderator analysis will be conducted” but there is no discussion of this further in the main text. Which variables will the authors investigate as moderators and mediators?
Response 12: At the end of Introduction, we included the variables that will be investigated as moderators and mediators. It now reads:
“It is expected that the blended CBT intervention will be as effective as treatment usually provided for PPD in decreasing depressive symptoms. In this study, we will evaluate the mediating role of psychological competences (self-compassion, emotion regulation, psychological flexibility) in treatment response. These mechanisms have been core psychological processes underlying the development of the Be a Mom program [31]. Moreover, previous studies have found that these psychological mechanisms were associated with improvements in depressive symptoms in the perinatal period [30, 32, 33]. We will also examine the moderator effect of characteristics of the patient (e.g., sociodemographic characteristics, motivation for therapy) and of the therapeutic process (e.g., therapeutic relationship, user’s satisfaction) in the efficacy of the blended intervention for PPD.” (see pages 2 -3)
We also clarified that mediating and moderating analysis will be conducted in Methods section. (see page 10)
Minor edits
Point 13: Line 306, please change to “existing”
Response 13: Thank you for noticing, we have corrected this.

Reviewer 2 Report
Thank you for the opportunity to review this manuscript.
I have some considerations for the authors:
Complete the Introduction section with more recent references.
Use more operational verbs to formulate objectives.
Research hypotheses should be included at the end of the introductory section, and they should be clearly stated. This will later make it possible to contrast with the results obtained in the Discussion section.
They must provide data on the reliability of the instruments applied for data collection.
In the Conclusions, they say: "This is the first study to develop and test the effectiveness of a blended CBT intervention for the treatment of PPD in the Portuguese context". Where are the Results of the application of the program? If it is a question of the approach of an intervention program and no results are presented, this should be clear throughout the entire manuscript (also in the title). Please review the structure and contents of the entire document to make it clear from the beginning what exactly the study consists of. It is confusing in the current version there are some statements that create expectations for the reader to find a results section of the application. If there is a pilot application, please include the results in the corresponding section.
Author Response
Response to Reviewer 2 Comments
Point 1: Complete the Introduction section with more recent references.
Use more operational verbs to formulate objectives.
Response 1: As suggested, we added more recent references (from 2020; cf. References) and used operational verbs (e.g., to examine, to evaluate) to formulate the study objectives. (see pages 2 - 3)
Point 2: Research hypotheses should be included at the end of the introductory section, and they should be clearly stated. This will later make it possible to contrast with the results obtained in the Discussion section.
Response 2: We have clarified the objectives and hypotheses, it now reads:
“Therefore, we herein outline the protocol for a randomized controlled trial to examine the acceptability and efficacy of a blended CBT intervention for PPD treatment, considering post-intervention and follow-up improvements in primary and secondary outcomes. It is expected that the blended CBT intervention will be as effective as treatment usually provided for PPD in decreasing depressive symptoms. In this study, we will evaluate the mediating role of psychological competences (self-compassion, emotion regulation, psychological flexibility) in treatment response. These mechanisms have been core psychological processes underlying the development of the Be a Mom program [31]. Moreover, previous studies have found that these psychological mechanisms were associated with improvements in depressive symptoms in the perinatal period [30, 32, 33]. We will also examine the moderator effect of characteristics of the patient (e.g., sociodemographic characteristics, motivation for therapy) and of the therapeutic process (e.g., therapeutic relationship, user’s satisfaction) in the efficacy of the blended intervention for PPD.” (see pages 2 - 3)
Point 3: They must provide data on the reliability of the instruments applied for data collection.
Response 3: As suggested by the reviewer, we added information about of the psychometric properties of the Portuguese versions of the instruments presented and that will be used in this study. (see pages 8 -9)
Point 4: In the Conclusions, they say: "This is the first study to develop and test the effectiveness of a blended CBT intervention for the treatment of PPD in the Portuguese context". Where are the Results of the application of the program? If it is a question of the approach of an intervention program and no results are presented, this should be clear throughout the entire manuscript (also in the title). Please review the structure and contents of the entire document to make it clear from the beginning what exactly the study consists of. It is confusing in the current version there are some statements that create expectations for the reader to find a results section of the application. If there is a pilot application, please include the results in the corresponding section.
Response 4: This article only presents the study protocol for the blended CBT intervention.
The publication of the protocol for a randomized controlled trial helps to follow the best standards in the study conduction and to monitor eventual changes made during the trial. It also allows to inform the scientific community of the development of a new form of intervention, in particular a blended CBT intervention for the treatment of postpartum depression, and therefore can avoid duplication of research.
Since it was not clear that this paper reported a study protocol, we have made some changes:
Title: “A blended cognitive-behavioral intervention for the treatment of postpartum depression: Study protocol for a randomized controlled trial”
Introduction:
- “This article presents the study protocol for a blended CBT intervention combining face-to-face sessions with the online program Be a Mom for the treatment of PPD in the Portuguese context.” (see page 2)
- “Therefore, we herein outline the protocol for a randomized controlled trial to examine the acceptability and efficacy of a blended CBT intervention for PPD treatment” (see page 2)
Throughout the manuscript, we used the term ‘study protocol’ and used future verbal tenses.
A pilot study will be conducted but there are still no results of its application. We also added this information:
“A pilot study with women with a clinical diagnosis of PPD will be conducted prior to the RCT, to assess the acceptability and feasibility of the structure and content of the blended intervention, and to gather preliminary evidence of its clinical efficacy (non-controlled). Appropriate adjustments to the blended intervention protocol will be done accordingly.” (see page 4)

Round 2
Reviewer 2 Report
Thanks for the clarifications and improvements made to the manuscript.